# Effect of specificity of health expenditure questions in the measurement of out-of-pocket health expenditure: evidence from field experimental study in Ghana

Isaiah Awintuen Agorinya [1,2,3] Amanda Ross,[2,3] Gabriela Flores,[4] Tessa TanTorres Edejer,[4] Maxwell Ayindenaba Dalaba [5,6] Nathan Kumasenu Mensah,[7] Lan Le My [2,3] Yadeta Dessie,[8] Jemima Sumboh,[5] Abraham Rexford Oduro,[5] J Akazili,[9] Fabrizio Tediosi[2,3]

**Correspondence to**
Dr Isaiah Awintuen Agorinya;
iagorinya@gmail.com

## ABSTRACT

**Background** The effect of number of health items on out-of-pockets (OOPs) has been identified as a source of bias in measuring OOPs. Evidence comes mostly from cross-sectional comparison of different survey instruments to collect data on OOPs. Very few studies have attempted to validate these questionnaires, or distinguish bias arising from the comprehensiveness of the OOPs list versus specificity of OOPs questions.

**Objectives** This study aims to estimate biases arising from the specificity of OOPs questions by comparing provider and household's information.

**Methods** A generic questionnaire to collect data on household's OOPs was developed following the nomenclature proposed in division 6 of the classification of household final consumption 2018. The four categories within such division are used to set the comprehensiveness of the OOPs list, the specificity within each category was tailored to the design of the nationally representative living standard survey in Ghana where a field experiment was conducted to test the validity of different versions. Households were randomised to 11, 44 or 56 health items. Using data from provider records as the gold standard, we compared the mean positive OOPs, and estimated the mean ratio and variability in the ratio of household expenditures to provider data for the individual households using the Bland-Altman method of assessing agreement.

**Findings** We found evidence of a difference in the overall mean ratio in the specificity for OOPs in inpatient care and medications. Within each of these two categories, a more detailed disaggregation yielded lower OOPs estimates than less detailed ones. The level of agreement between household and provider OOPs also decreased with increasing specificity of health items.

**Conclusion** Our findings suggest that, for inpatient care and medications, systematically decomposing OOPs categories into finer subclasses tend to produce lower OOPs estimates. Less detailed items produced more accurate and reliable OOPs estimates in the context of a rural setting.

### Strengths and limitations of this study

► The main strength of this study is the use of provider data as a 'gold standard' to validate self-reported household out-of-pocket (OOP) health expenditures in a rural African setting.

► This study has quantified the level of agreement and variability in current household survey instruments that uses different levels of health expenditure items to estimate OOPs.

► The study validated spending categories contained in the latest classification of household final consumption revisions 2018, which has previously not been done

► Generation of provider data posed as a major limitation to this study as this was not a routine practice by health providers.

► Sample sizes for each spending category was largely affected by the fraction of households reported OOPs that accurately matched with provider data

## INTRODUCTION

Household out-of-pocket health (OOP) payments are direct payments for services from households' primary income or savings with no third-party payer involved.[1] OOPs are an important measure of performance of the health financing system. They are used to monitor to what extent voluntary and unpredictable payments are used to mobilise money within the health system and the impact of such payments on the household's living standard and ability to spend on other basic needs[2–6]

An important source of information to track OOPs are household surveys, especially in countries where much private healthcare financing occurs without the generation of

linked, reliable and comprehensive routine data. Household consumption and expenditure surveys, household utilisation and expenditure surveys, and health surveys with information on health expenditures can be used to gather data on OOPs.

These surveys are not standardised as they can differ, among other things, in the comprehensiveness (the number of main categories covered by the survey) as well as the level of detail (the specificity within each main category of health product or service).[7–12] Lu et al[8] comparing estimates from 50 countries using World Health Survey data reported that—if there was one rather than eight health categories, then the average reported health spending tended to be lower. The number of questions varies: Heijink et al[11] reviewed survey questionnaires for 114 countries and found that the number of health expenditure questions mostly vary between 1 and 25, with some outliers falling over this range.[11] Lavado et al[13], also found the number of questions on health expenditure to range from 1 to 274 in 214 surveys and estimated that an additional question on OOPs increases health expenditure share by 1%.[13] However, neither Lavado et al[13], nor Heijink et al[11] distinguished bias arising from comprehensiveness of the OOPs list versus specificity within each category.

To the best of our knowledge, no validation study has so far been conducted to assess the effect of the specificity of the expenditure questions on the accuracy and reliability of reported OOPs in the context of low-middle income countries. In this paper, we compare household responses on health expenditure questions to the corresponding health provider data in order to assess and quantify agreement. We use questionnaires with three different levels of detail to assess the effect of the specificity of health items on the accuracy of reported OOPs

## METHODS
### Strategy
The overall project, in which this study is nested, assesses the impact of different survey characteristics, such as specificity, recall period, and whether the survey is door to door or telephone based, on the accuracy of household-reported OOP payments (iHOPE). It was implemented in three health and demographic surveillance sites (HDSS) of the INDEPTH-Network located in northern Ghana (Navrongo HDSS, mostly rural), Burkina Faso (Ouagadougou demographic surveillance site, informal urban setting) and Vietnam (FilaBavi demographic surveillance site, mixed rural/urban setting).

The INDEPTH-Network platform provided the project with the opportunity to identify and track households and link them to health provider records to be able to validate household reported OOP expenditures.

In this paper, we report the study on the effect of the specificity of health expenditure categories on reported OOPs conducted in Navrongo HDSS (Ghana). For this experiment, three versions of health expenditure

modules with different level of specificity of health expenditure categories were developed and adapted to the structure of the Ghana Living Standards Survey 6 instrument.[14] These three versions were all comprehensive in that they all captured the major healthcare consumption groups that constitute the main categories of healthcare expenditures individuals are confronted with as identified by the classification of household final consumption (COICOP), 2018 version.[15] They differed in the level of detail (specificity) within each class. Health expenditures were first compared across versions without any gold standard. This is what most studies to date have been able to analyse. The value added of this study is the use of health provider records to identify the level of agreement between two different sources (provider records vs household reports) of the same health expenditures made by household members, and to compare the level of agreement between the different questionnaire versions.

### Study settings
This study was implemented at the Navrongo HDSS site located in the Northern region of Ghana. The site has two administrative districts with an estimated total population of 160 000. The site has one public hospital, one health research centre, one private clinic, seven health centres and 27 community-based health compounds. A number of pharmacy shops, chemical and drug shops, petty traders and peddlers, herbalists, faith-based and traditional healers also operate in the area. The research centre collects vital socio-demographic data every 4 months while household characteristics and assets are collected every 2 years.[16]

### Study context: Ghana's National Health Insurance Scheme
Ghana enacted legislation (National Health Insurance Act 2003 (Act 650) and implemented a National Health Insurance Scheme (NHIS) aimed at reducing out of pocket payments that were estimated to account for up to 48% of the total health expenditure.[17] The NHIS offers free access to a package of diagnostic, inpatient and outpatient services covering 95% of conditions afflicting Ghanaians with services including primary curative care to care at tertiary facilities but challenges within the operationalisation of the scheme are thought to expose subscribers to OOPs.[18] The NHIS is intended to cover the whole population with formal workers automatically covered through deductible social contributions, informal workers registered through annual premium payments while vulnerable people are exempted from paying premiums.

The scheme is largely financed through tax and a small proportion from contributions and donations. In 2014, the scheme covered only about 40% of Ghana's population (10.5 million active subscribers) with about 69% of these exempted from any form of payment to the scheme.[19] The exempt groups include indigent people, pregnant women and very poor households covered by a social intervention programme called Livelihood

Empowerment Against Poverty. Household members who enrol into the Ghana NHIS are assured of free services within the scope of the NHIS beneficiary package. However, subscribers to the NHIS may be exposed to OOPs when accessing health providers for medicines, laboratory tests and other consumables which may not be available at the provider due for example to stockouts[18] or non-accredited NHIS provider. The uninsured population are required to pay OOP to be able to access healthcare.[19]

In Ghana OOPs are mostly incurred by households that are not registered with the NHIS, for services that are not included in the NHIS benefit package, or due to challenges within the operationalisation of the health insurance scheme or households accessing healthcare from a non-health insurance accredited private health provider. Informal payments for healthcare are not common in the study area (this was investigated during piloting) as seen in other areas and therefore has are not included in this study.

### Study design

This study uses health provider records as the 'gold standard' to compare to household reported OOPs, recognising that the provider records are not a perfect gold standard in the absence of a pre-existing formal recording system in place. Two sets of data were collected in this study. The first set of data was captured from households in a cross-sectional field survey conducted between May 2017 and December 2017 and the second set of data was obtained from health provider records within the same period.

Households were randomised to one of three versions of a household questionnaire on consumption expenditure for face-to-face interview. All three versions of the questionnaire were fielded during the same time period and included questions on OOP health spending for inpatient care services, preventive care services, other outpatient care services, other health services, medicines and health products. They however differed in the level of specificity within each one of these main OOPs categories. The versions were labelled; V.1 for the questionnaire with 11 items covering the 6 health categories previously mentioned V.2 and V.3 for the questionnaire versions with a maximum of 44 and 56 items, respectively (see figure 1 and table 1 in online supplemental material 1). All three versions of the instrument used similar recall periods of 4 weeks for items listed under the categories medicines, other outpatient services, other health services; 12 months for inpatient care services and other health products; and 6 months for OOPs related to preventive care services. Online supplemental material 1 discusses in detail how the structure of the questionnaire was obtained. The specificity of OOPs categories is also available from online supplemental material 2.

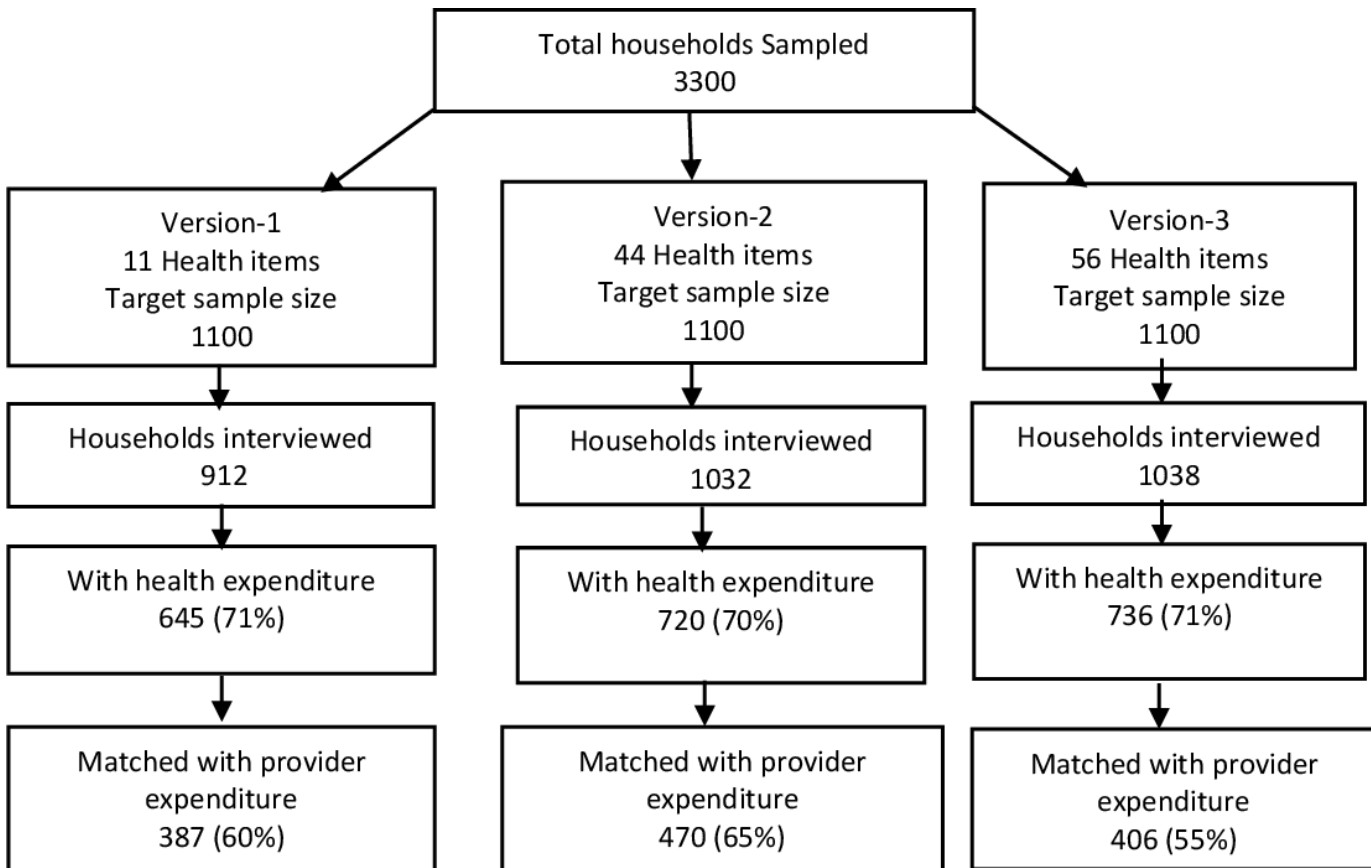

**Figure 1** Flow chart of household enrolment and matching to provider records.

**Table 1** Demographic characteristics of household and household head

| | Questionnaire V.1 | | Questionnaire V.2 | | Questionnaire V.3 | | All questionnaire versions combined | |
|---|---|---|---|---|---|---|---|---|
| Total no of households | N=925 | | N=1062 | | N=1036 | | N=3023 | |
| | n | % | n | % | n | % | n | % |
| Gender | | | | | | | | |
| Male | 582 | 63 | 705 | 66 | 647 | 62 | 1934 | 64 |
| Marital status | | | | | | | | |
| Married | 566 | 61 | 687 | 65 | 606 | 59 | 1859 | 62 |
| Level of Education | | | | | | | | |
| No education | 451 | 49 | 574 | 54 | 572 | 55 | 1597 | 53 |
| Primary | 200 | 22 | 214 | 20 | 202 | 20 | 616 | 20 |
| Junior high school | 144 | 16 | 141 | 13 | 136 | 13 | 421 | 14 |
| Senior high school | 44 | 5 | 51 | 5 | 49 | 5 | 144 | 5 |
| Vocational/technical/college/graduate | 86 | 9 | 82 | 8 | 77 | 7 | 245 | 8 |
| Religion | | | | | | | | |
| Christians | 519 | 56 | 501 | 47 | 571 | 55 | 1591 | 53 |
| Islam | 54 | 6 | 148 | 14 | 41 | 4 | 243 | 8 |
| Traditional | 314 | 34 | 355 | 33 | 362 | 35 | 1031 | 34 |
| No religion | 38 | 4 | 58 | 6 | 62 | 6 | 158 | 5 |
| Age group | | | | | | | | |
| 15–19 | 40 | 4 | 56 | 5 | 43 | 4 | 139 | 5 |
| 20–34 | 51 | 6 | 77 | 7 | 67 | 7 | 195 | 6 |
| 35–64 | 572 | 62 | 621 | 59 | 589 | 57 | 1782 | 59 |
| 65+ | 262 | 28 | 308 | 29 | 337 | 33 | 907 | 30 |
| Mean age (SD) | 55 | 17 | 54 | 17 | 56 | 17 | 55 | 17 |
| Household size | | | | | | | | |
| 1 person | 67 | 7 | 82 | 8 | 64 | 6 | 213 | 7 |
| 2–5 persons | 418 | 45 | 549 | 52 | 566 | 55 | 1533 | 51 |
| 6 and above | 441 | 48 | 432 | 41 | 403 | 39 | 1276 | 42 |

We created a database of provider records to validate the household reported OOPs. Both private and public healthcare providers within the study area were engaged by the project team to extract records covering a 13-month period which in some cases (mostly for private providers) also required improving the recording of provider records. Health expenditures reported by any member of the household were tracked and a corresponding health provider record obtained to create a matched sample for validation. A detailed description of the matching procedure can be seen in online supplemental material one under the section matching. Matched sample in this study refers to one-on-one paired records households reported OOPs to heath provider recorded OOPs for the same reason of incurring the health cost. Unmatched households in this study also refers to all OOPs reported by households without pairing such expenditures to corresponding health provider records. Household heads were the most common respondents for the household survey but in some cases, other individuals within the household were nominated by the household head to provide responses.

## Study population and sampling

All households registered under the Navrongo demographic surveillance site database constituted our study population. The households were randomly selected. The sample size for the household cross-sectional survey was based on estimating the agreement between two quantitative measurements. We followed the Bland and Altman approach which suggests a rule of thumb that 100–200 observations is adequate for sufficient precision when assessing agreement.[20] Based on unpublished district health management report in the study area, the probability of spending on outpatient for a 2-week recall period was 15.5% and 10% for a 12-month recall for inpatient care. To achieve a minimum of 100 households with reported inpatient care (outpatient spending is included in this sample), the sample size required would be 1000 households plus 10% non-response rate which gives a total of 1100 households per questionnaire version, and 3300 households for all three versions.

## Data analysis

We carried out separate analyses for the unmatched and matched data to assess the effect of the number of health items on the amount and accuracy of the reported OOPs. We assessed the consistency of the findings in these two analyses. For the unmatched samples, we summarised the OOPs by spending categories and household characteristics for each questionnaire version using arithmetic means and SD for continuous variables and proportions for categorical variables. Summarising and comparing household reported OOPs are consistent with the methodological approach used in most studies to date without information from the provider.[11] We summarised the effect of number of items on mean reported OOPs by spending category using the ratio of each category to the reference version (V.1 which had 11 items). The confidence intervals for the ratio were calculated using the bootstrapping method. Only four categories of spending were considered instead of six due to the small number of households reporting any OOPs for health products and other health services.

For the matched samples, we adopted the Bland-Altman approach[21 22] to assess the level of agreement between household reported OOPs and corresponding provider records. We first estimated the agreement for each questionnaire version separately. We calculated the ratio of individual household to matched provider OOPs. The ratio was preferred to the simple difference since it had a reasonably constant distribution over the range of the provider OOPs, whereas the difference between household and provider OOPs was dependent on the level of provider OOPs. The log-transformation of the ratio was used for analysis due to the skewed distribution of the ratios.[21 22] For each questionnaire version, we estimated the overall mean ratio using the geometric mean of the ratios and the variability using the 95% limits of agreement (LoA) (ie, limits within which 95% of the ratios are expected to lie). To compare the three versions of the questionnaire and assess the effect of number of health items on OOPs, we compared the overall mean ratio and variability using regression models proposed by Bland-Altman.[22] Specifically, for the mean ratio, we fitted a regression model with the log ratio of household to provider OOPs as the outcome variable and questionnaire version as an explanatory variable. This provides an estimate of the effect of the questionnaire version on the bias, measured by the mean log ratio with corresponding CI and p value. We included a random effect parameter to account for the clustering of the households within clusters defined by the Navrongo HDSS. For the effect of the questionnaire version on the variability, a second regression model was used with the outcome of the absolute values of the residuals previously obtained against the questionnaire version. Since the dependent variable in the first regression is on the log scale, all results were transformed to the ratio scale for ease of interpretation. Questionnaire V.1 which has the least number of health items (11 items) was used as the reference group for the

regression models. Data were analysed using STATA V.14 (StataCorp).

## Consent to participate

Informed consent was obtained from all study participants.

## Patient and public involvement

Per the design and objective of this study, it was not appropriate to involve the public in the design, or conduct, or reporting, or dissemination plans of our research.

## RESULTS

### Sample distribution and matching

A total of 3300 households were sampled and randomised to receive one of the three questionnaire versions. The response rate was 83% in version 1, 94% in version 2% and 94% in version 3. Approximately 70% of households reported any form of OOP across all three questionnaire versions as expected. In terms of the matching of household expenditures to provider data, the proportion of households that matched was 60% in version 1, 65% in version 2% and 55% in version 3. Figure 1 shows a flow chart that summarises the sample distribution, the proportion of households incurring OOPs and the overall proportion of OOPs that matched with provider records in each questionnaire version.

### Sociodemographic characteristics of households

A total of 3023 households were interviewed out of 3300 sampled. Of these, 64% of the household heads were males, the majority (89%) of the household heads were above 34 years of age, 62% of the household heads were married, slightly more than half (53%) of household heads did not have any formal education or were Christians and the mean age was 55 years (17 SD). The majority of households had more than one member with 51% having 2–5, and 42% with 6 or more (table 1). From online supplemental table 1 in online supplemental material 3, the distribution of demographic characteristics was similar between households in the three questionnaire versions, households with health expenditure vs those without, and households with matched provider records versus households without successful matched provider records. (online supplemental material 3: online supplemental table 1).

### Proportion reporting any OOPs by spending category

Overall, 2969 households provided information about household health spending of which 71% of households reported any form of OOPs. The proportion of all households reporting OOPs for inpatient care over the 12 month recall period was 18%, for preventive care over the past 6-month recall period 9%, for outpatient care within the past 4 weeks 12%, and slightly more than half (56%) of the households also reported having incurred OOPs on medications over the past 4 weeks regardless of the number of health items (table 2).

**Table 2**  Health expenditures—proportion of households reporting positive OOPs by spending category

| No of households with positive health by category: new COICOP classification | Questionnaire V.1 N=901 | | | Questionnaire V.2 N=1032 | | | Questionnaire V.3 N=1036 | | |
|---|---|---|---|---|---|---|---|---|---|
| | No of health items | n | % | No of health items | n | % | No of health items | N | % |
| Inpatient care services | 2 | 170 | 19 | 14 | 177 | 17 | 14 | 193 | 19 |
| Preventive services | 2 | 137 | 15 | 5 | 92 | 9 | 5 | 46 | 4 |
| Other health services | 1 | 9 | 1 | 2 | 5 | 0.5 | 2 | 2 | 0.2 |
| Outpatient | 2 | 81 | 9 | 12 | 181 | 18 | 12 | 105 | 10 |
| Medicines | 2 | 487 | 54 | 9 | 560 | 54 | 16 | 609 | 59 |
| Health products | 2 | 36 | 4 | 2 | 25 | 2 | 7 | 18 | 2 |
| No of households with any health expenditure | 11 | 645 | 71 | 44 | 720 | 70 | 56 | 736 | 71 |

COICOP, classification of household final consumption; OOP, out-of-pocket.

### Mean OOPs reported by households by spending category (unmatched analysis)

The mean household OOPs tended to be larger for lower numbers of health expenditure items (less specificity) in all main spending categories except health products where there are few households with spending, and inpatient care where there was no consistent pattern with increasing specificity. The differences in specificity between questionnaire versions were significant for only preventive care and medicines (table 3). Online supplemental tables 2 and 3 in online supplemental material 3 also shows the summary of mean OOPs compared for matched and unmatched households versus provider estimates across the three versions and by spending categories. The tables show that, the average OOPs for the matched households is similar to the average OOPs of the unmatched households for inpatient care, medicines and outpatient care.

### Quantifying the level of agreement and variability between household and provider OOPs (matched analysis)

Overall, the household OOPs tend to be higher than the corresponding provider OOPs. We assessed the agreement between the matched household and provider OOPs first for each questionnaire version separately by estimating the overall level of agreement and variability for each type of health expenditure category. The overall bias, measured as the geometric mean ratio of household to provider OOPs saw a increasing trend with increasing numbers of health items for OOPs in only inpatient care and medicines (table 4). There was, however, no evidence of an effect for outpatient or preventive care. This evidence suggests that disaggregating health expenditure items into finer specific items tended to decrease the level of agreement between matched household OOPs and corresponding provider records for inpatient care and medicines as observed in the increasing trend in the mean bias. Tables 2 and 3 in online supplemental material 1 shows details of the levels of matching for different

types of providers and services received. The combine matching rate was 59%.

We investigated why the unmatched analysis pointed to lower mean OOPs with increasing numbers of items, but the matched analysis to higher mean OOPs. There were variations in the proportions of households with matched records across type of service received by household members and type of health providers. OOPs incurred from diagnostics, the pharmacy and community health and family planning service facilities patients tended to match better than in the hospitals as observed in table 2 of online supplemental material 1. The demographic characteristics of households were similar for all households with OOPs and the matched households only (online supplemental material 3 and table 1). The mean OOPs within spending categories tended to be lower for the matched households (online supplemental material 3 and table 2). The moderate matching rates, and a tendency for higher household-reported OOPs to be omitted, suggests that the matching process may have influenced the matched analysis results.

### DISCUSSION

We present evidence of the influence of the specificity of health categories on the level of agreement and variability of reported household positive OOPs by spending category. This is done by comparing household and provider data. In this study, we found that the unmatched analysis (where household data is not compared with provider data), comparing mean reported OOPs for all households, suggested a tendency for a higher number of items to result in lower positive OOPs for medicines and preventive services. This suggests that being more detailed as a consequence of increasing the number of health items within each category leads to lower mean positive OOPs in these spending categories. In assessing the agreement between matched household-reported and provider OOPs, our findings suggests that, the level of agreement

**Table 3** Arithmetic mean OOPs by health category and questionnaire version

| Spending category | Questionnaire V.1<br>11 disaggregated health items | | | | Questionnaire V.2<br>44 disaggregated health items | | | | | Questionnaire V.3<br>56 disaggregated health items | | | | |
|---|---|---|---|---|---|---|---|---|---|---|---|---|---|---|
| | No of health items aggregated | N | household average OOPs (Ghc) | SD | Number of health items aggregated | N | household average OOPs (Ghc) | SD | Estimated ratio of the means (V.2/V.1) (95% CI) | Number of health items aggregated | N | household average OOPs (Ghc) | SD | Estimated ratio of the means (V.3/V.1) (95% CI) |
| Outpatient | 2 | 81 | 64 | 135 | 12 | 181 | 43 | 130 | 0.70 (0.20 to 1.21) | 12 | 105 | 44 | 78 | 0.75 (0.27 to 1.22) |
| Inpatient | 2 | 171 | 319 | 527 | 14 | 177 | 398 | 809 | 1.25 (0.75 to 1.74) | 14 | 193 | 287 | 716 | 0.92 (0.51 to 1.34) |
| Medicines | 2 | 487 | 41 | 140 | 9 | 560 | 29 | 78 | 0.71 (0.44 to 0.98) | 16 | 609 | 29 | 76 | 0.66 (0.44 to 0.88) |
| Preventive care | 2 | 137 | 59 | 95 | 5 | 92 | 34 | 53 | 0.60 (0.33 to 0.87) | 5 | 46 | 31 | 44 | 0.57 (0.27 to 0.88) |
| Other medical services | 1 | 8 | 203 | 201 | 2 | 5 | 113 | 217 | 0.56 (-) | 2 | 2 | 12 | 4 | 0.06 (-) |
| Health products | 2 | 36 | 71 | 133 | 2 | 25 | 160 | 250 | 2.38 (-) | 7 | 18 | 165 | 232 | 2.32 (-) |

Note: the currency used is the GHC. US GHC4.2 was equivalent to US$1 at the time of collecting data.
GHC, Ghana cedi; OOP, out-of-pocket.

between provider and household OOPs decreased with increasing specificity for medicines and inpatient care. These trends maybe attributed to the inability of households to either recall the specific name of the medicine/service received from the health provider or they did not know the name of the medicine/service at all when such information is required at a very detailed level. Another potential factor could be a possible introduction of bias by the matching process in our study, with the possibility that smaller amounts matched better with provider data than larger amounts. These factors may require further investigations beyond the scope of this current study.

A number of studies have investigated and documented the potential effect and consequence of varying number of health items on the estimation of OOPs using nationally representative survey data.[8 12 13 23] All these studies have mostly focused their investigations on the effect of number of health items on total household health expenditure without examining the effect by spending categories or distinguishing between comprehensiveness of the health expenditure list (number of health categories covered) from specificity of each category (number of questions per health category) therefore making such studies deficient in identifying the most reliable and accurate survey tools.

The primary purpose of this study is to investigate the effect of the specificity of the health expenditure question on both the total amount spent on health OOP and amounts spent by type of service or medical products. While the total amount is a critically needed to assess to what extent countries rely on households' direct contributions to fund the health system within the national health account framework on the one hand and their impact on household's welfare in the context of financial protection monitoring, information on levels by type of spending categories are critical to better inform policies. It is important to know if households are mostly contributing to fund medicines vs inpatient care to give just one example. Hence, knowing that the total level of OOPs is over or under-estimated is insufficient and understanding which type of expenditure is driving such over or under-estimation is critical.

This paper is the first one to have attempted to shed light on this particular aspect. Others have focused on either one specific category (eg, inpatient care) or completely ignored this issue[13] in most cases relying on cross-sectional comparisons across multiple countries that also differ in terms of the choice of the recall period and total number of non-medical questions (4, 11–13).

By using an experimental design, we were able to control for other potential confounders (recall period, number of non-medical expenditure question, type of survey). We also aimed at validating households' responses. The challenges faced during the implementation impacted in our initial plan in two ways: first we are not able to validate the total amount spent on health OOP on all types of good and services. Second, we had to restrict the validation part to those households that reported a health expenditure

**Table 4** Agreement between households and provider in OOPs by number of health expenditure items and spending category

| No of health items within each spending category | No of households | Total no of health questions | Mean ratio | 95% limits of agreement of mean ratio | Estimated difference in mean ratio between questionnaire versions and CI and p value | Estimated difference in SD of mean ratio between questionnaire versions and CI and p value |
|---|---|---|---|---|---|---|
| Outpatient care | | | | | p=0.49 | p=0.50 |
| 2 health items | 44 | 11 | 1.02 | 0.05 to 21.2 | – | – |
| 12 health items | 126 | 44 | 1.21 | 0.05 to 26.7 | 1.29 (0.74 to 2.23) | 1.02 (0.73 to 1.42) |
| 12 health items | 47 | 56 | 1.56 | 0.12 to 19.6 | 1.47 (0.77 to 2.80) | 0.84 (0.57 to 1.25) |
| Inpatient care | | | | | p=0.003 | p=0.01 |
| 2 health items | 91 | 11 | 3.88 | 0.17 to 86.2 | – | – |
| 14 health items | 99 | 44 | 6.61 | 0.16 to 270.7 | 1.63 (0.99 to 2.69) | 1.35 (1.03 to 1.76) |
| 14 health items | 100 | 56 | 9.19 | 0.51 to 161.2 | 2.34 (1.44 to 3.83) | 0.93 (0.71–1.21) |
| Medicines | | | | | p=0.023 | p=0.33 |
| 2 health items | 302 | 11 | 1.26 | 0.10 to 16.0 | – | – |
| 9 health items | 381 | 44 | 1.35 | 0.09 to 19.5 | 1.14 (0.91 to 1.42) | 1.04 (0.91 to 1.19) |
| 16 health items | 354 | 56 | 1.62 | 0.18 to 15.0 | 1.36 (1.01 to 1.70) | 0.95 (0.83 to 1.08) |
| Preventive care | | | | | p=0.290 | p=0.51 |
| 2 health items | 86 | 11 | 1.21 | 0.09 to 15.15 | – | – |
| 5 health items | 67 | 44 | 0.89 | 0.04 to 18.9 | 0.74 (0.46 to 1.14) | 1.19 (0.88 to 1.62) |
| 5 health items | 22 | 56 | 1.33 | 0.08 to 22.4 | 1.07 (0.55 to 2.05) | 1.04 (0.68 to 1.60) |

Note: The unit of the estimated difference is the mean ratio expressed as a ratio between household and provider for each questionnaire version.
OOP, out-of-pocket.

(the spenders). Despite these limitations, this paper remains the first one to attempt validation on multiple type of spending and we are able to show that the accuracy of different type of services/products is differently affected

## CONCLUSION

This study describes a novel validation study in a rural setting that operates a demographic surveillance system. The evidence from this study suggests caution in interpreting OOPs from different survey instruments relying on different levels of detail per spending category even in the absence of differences in recall period and other survey questionnaire design features. We found that systematically decomposing health expenditure items into more specific and finer subclasses leads to lower average OOPs for outpatient care, medicines and preventive care when comparing across different version without comparing to provider records. Validation studies need to consider the possibility of bias introduced by the matching approaches.

**Author affiliations**
[1]Epidemiology and Biostatistics, University of Health and Allied Sciences, Hohoe, Ghana
[2]Epidemiology and Public Health, Swiss Tropical and Public Health Institute, Basel, Switzerland
[3]University of Basel, Basel, Switzerland
[4]World Health Organization, Geneve, Switzerland
[5]Navrongo Health Research Centre, Navrongo, Ghana
[6]Institute of Health Research, University of Health and Allied Sciences, Ho, Volta Region, Ghana
[7]Department of Health Information Management, University of Cape Coast, Cape Coast, Ghana
[8]Public Health, Haramaya University, Harar, Ethiopia
[9]Ghana Health Service, Accra, Ghana

**Acknowledgements** The authors wish to thank all the study participants, health facilities and the entire iHOPE Ghana team for participating in this study. We are very grateful to INDEPTH-Network and Swiss Topical and Public Health Institute for their technical support during the entire phase of this study. We also appreciate the financial support from the Bill and Melinda Gates foundation for the study.

**Contributors** The research protocol was developed by TTTE, GF, JA and FT. The research protocol was written by GF, TTTE, JA, FT and YD. The questionnaires were developed by GF and IAA. The analytical plan was developed by AR, IAA, LLM and GF. Data collection: IAA, MAD, NKM and ARO. Data cleaning: IAA, NKM, MAD and JS. Data analysis: IAA and ARO. First draft was written by: IAA and ARO. All authors contributed to writing of the manuscript.

**Funding** This project was jointly funded by the INDEPTH-Network in Accra and the Swiss tropical and public health institute of the University Of Basel, Switzerland through a grand from Bill and Melinda gates foundation, grant number OPP1113162. GF, TTTE and ARO were partially supported by WHO.

**Competing interests** None declared.

**Patient consent for publication**  Not required.

**Ethics approval**  The Ethical Review Board of the Navrongo Health Research Centre, Ghana (NHRCIRB217) approved for the conduct of the study.

**Provenance and peer review**  Not commissioned; externally peer reviewed.

**Data availability statement**  Data are available in a public, open access repository. Extra data can be accessed via the Dryad data repository at http://datadryad.org/ with the doi:10.5061/dryad.nk98sf7sw.

**ORCID iDs**

Isaiah Awintuen Agorinya http://orcid.org/0000-0002-6194-2021
Maxwell Ayindenaba Dalaba http://orcid.org/0000-0002-7101-769X
Lan Le My http://orcid.org/0000-0003-1887-0752

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
