## [Reviewer comments · BMJ Open]

ARTICLE DETAILS

TITLE (PROVISIONAL)	Effect of specificity of health expenditure questions in the measurement of out-of-pocket health expenditure: Evidence from field experimental study in Ghana
AUTHORS	Agorinya, Isaiah; Ross, Amanda; Flores, Gabriela; TanTorres Edejer, Tessa; Dalaba, Maxwell Ayindenaba; Mensah, Nathan; Le My, Lan; Dessie, Yadeta; Sumboh, Jemima; Oduro, Abraham; Akazili, J; Tediosi, Fabrizio

VERSION 1 – REVIEW

REVIEWER	Pablo Villalobos Dintrans Universidad de Santiago, Chile
REVIEW RETURNED	01-Aug-2020

GENERAL COMMENTS	The article proposes an interesting and much needed discussion on the way we measure OOP health expenditures. It does a good job explaining the problem and motivating the need for the research question. It also provides a good explanation of the methods used. Main comments: * A more detailed discussion on the features of the health system and how they can influence the study design and results is required* More information to assess the success of the randomization process is needed* Provide details on the methodological choices made in constructing the different surveys* Since one of the main methodological issues (matching) presented several problems, the article needs to convince the reader about its specific contribution. This requires a better discussion on how it compares to other articles in the same field. Please see attachment for more detailed comments. The reviewer provided a marked copy with additional comment. Please contact the publisher for full details.
---

REVIEWER	Sakthivel Selvaraj Public Health Foundation of India India
REVIEW RETURNED	30-Oct-2020

GENERAL COMMENTS	Title: Effect of specificity of health expenditure questions in the measurement of out-of-pocket health expenditure: Evidence from field experimental study in Ghana Manuscript ID: bmjopen-2020-042562
--

	The manuscript attempted to validate and assess the effect of the specificity of OOPs expenditure on the accuracy & reliability of OOPs estimates in LMICs. Such an exercise was made possible by the manuscript by comparing household responses to provider data. The manuscript is an important addition to existing literature on the design of survey tools in obtaining a robust and reliable OOP estimates. Key observations on Methods and Results are organized as follows: Page 5: Methods: The method section need to describe in detail all three versions of health care consumption goods and needs to distinguish the level of specificity between three versions. Since this is the core objective of the manuscript. Page 6: Under Study design, while the manuscript mentions all three questionnaires and the number of items captured but similar description of provider items captured also need to be outlined, for head-to-head comparison to make. Page 6: While sample size is outlined, how are these samples distributed across income and social groups, employment groups, insured, non-insured population, etc. must be described. The magnitude of OOP estimates and its variation are often influenced by these factors. Page 7: the section on Health Financing System in Ghana must figure in the introductory section for readers to appreciate the current health financing mechanism in Ghana. IN any case, health financing system section cannot be clubbed with methods section. Page 9: Ideally, the number of households interviewed should be considered as total sample rather than total sample identified as 1100 in each version. Or at least, please explain the reason for the difference between 1100 sample identified for each version and actual number of households interviewed. Also, it is not clear how many unmatched samples were drawn from the population and whether the sample size is robust needs to be described. Page 12-13: As reported in the manuscript in page 12, the overall bias increases with increasing numbers of health items for OOPs in inpatient care and medicines. But table 4 supports this claim only for inpatient care, while overall bias for medicines and outpatient appear similar. Mean bias in both outpatient and medicines expenditure appear to slightly rise but with increase in number of items for medicines while for outpatient the number of items remain the same. This require urgent attention and edits in the manuscript and the results need to be interpreted accordingly. The difference in mean bias in preventive care services with similar items (5 items) also require explanation. Specific points to be addressed in the manuscript: Page 4: Please expand COICOP, as it was introduced for the first time in the manuscript. Page 5: second line 'norther' should read as 'northern'.
--	--

	Page 7: Please check second line in Data Analysis section. The sentence has repeated twice 'reported OOPs' and 'reported OOPs'. Page 14: 5th Line: "preventing services" must be replaced by "preventive services". In last line 4 from bottom, 'provide' records must be replaced by 'provider' records.
--	---

VERSION 1 – AUTHOR RESPONSE

Reviewer: 1

Dr. Pablo Villalobos Dintrans, Universidad de Santiago de Chile Facultad de Ciencias Medicas

Comments to the Author:

The article proposes an interesting and much needed discussion on the way we measure OOP health expenditures.

It does a good job explaining the problem and motivating the need for the research question. it also provides a good explanation of the methods used.

Main comments:

Comment: A more detailed discussion on the features of the health system and how they can influence the study design and results is required

Response: This has been addressed in the manuscript now (Line 161)

Comment: More information to assess the success of the randomization process is needed

Response: We checked the summarized characteristics for households in each questionnaire group and found them to be similar. (Line 280 and Table 1). We do not use p-values to check randomization according to the CONSORT guidelines (Pocock et al, 2002; CONSORT 2010) so we relied on the summarized characteristics to assess group differences/similarities

Comment: Provide details on the methodological choices made in constructing the different surveys

Response: The estimation of the level of OOPs is also affected by the number of questions on non-health expenditure items. We, therefore, took into consideration the ratio of the number of health expenditure items to the number of non-health expenditure items before constructing different questionnaire versions (versions 1-3). The starting point was, to begin with the number of health items in the Ghana Living standards survey by aligning it to the revised COICOP 2018 structure for health items (division 6). This initial version is what we refer to as questionnaire version-1 in this study. Version 2 was derived by disaggregating version-1 further but following the ratio of health to non-health items. Version-2 was also derived in a similar way. The detailed structure of these question designs is also found in supplementary material 2. We have also included some text in the main manuscript for clarity (Line 216 - 240)

Comment: Since one of the main methodological issues (matching) presented several problems, the article needs to convince the reader about its specific contribution. This requires a better discussion on how it compares to other articles in the same field.

Response: This has now been included in the discussion section (Line 385 - 405)

- The primary purpose of this study is to investigate the effect of the specificity of the health expenditure question on both the total amount spent on health out-of-pocket and amounts spent by type of service or medical products. While the total amount is a critically needed to assess to what extent countries rely on households' direct contributions to fund the health system within the national health account framework on the one hand and their impact on household's welfare in the context of

financial protection monitoring, information on levels by type of spending categories are critical to better inform policies. It is important to know if households are mostly contributing to funding medicines versus inpatient care to give just one example. Hence, knowing that the total level of OOPs is over or under-estimated is insufficient and understanding which type of expenditure is driving such over or under-estimation is critical.

- This paper is the first one to have attempted to shed light on this particular aspect. Others have focused on either one specific category (e.g inpatient care) or completely ignored this issue (e.g. Lavado's paper) in most cases relying on cross-sectional comparisons across multiple countries that also differ in terms of the choice of the recall period and a total number of non-medical questions.
- By using an experimental design we were able to control for other potential confounders (recall period, number of non-medical expenditure question, type of survey). We also aimed at validating households' responses. The challenges faced during the implementation impacted in our initial plan in two ways: first we are not able to validate the total amount spent on health out-of-pocket on all types of good and services. Second, we had to restrict the validation part to those households that reported a health expenditure (the spenders). Despite these limitations, this paper remains the first one to attempt validation on multiple types of spending and we are able to show that the accuracy of different type of services/products is differently affected.

Reviewer: 2

Dr Sakthivel Selvaraj, Public Health Foundation of India

Comments to the Author:

Title: Effect of specificity of health expenditure questions in the measurement of out-of-pocket health expenditure: Evidence from field experimental study in Ghana

Manuscript ID: bmjopen-2020-042562

The manuscript attempted to validate and assess the effect of the specificity of OOPs expenditure on the accuracy & reliability of OOPs estimates in LMICs. Such an exercise was made possible by the manuscript by comparing household responses to provider data. The manuscript is an important addition to existing literature on the design of survey tools in obtaining a robust and reliable OOP estimates.

Key observations on Methods and Results are organized as follows:

Page 5: Methods: The method section need to describe in detail all three versions of health care consumption goods and needs to distinguish the level of specificity between three versions. Since this is the core objective of the manuscript.

Responds: We thank the reviewer for this important observation. We have now provided the detailed structure questionnaire design in the manuscript in addition to what is contained in supplementary material 2.

Comment: Page 6: Under Study design, while the manuscript mentions all three questionnaires and the number of items captured but similar description of provider items captured also need to be outlined, for head-to-head comparison to make.

Response: The design of the study was focused on household level reported health expenditures. These expenditures were then validated using data obtained from health providers where households incurred such expenditures. In this framework of the study design, respondents were asked to identify the health providers such expenses were made and the field team subsequently visited these facilities to extract the corresponding amounts recorded by the provider. Provider items were thus determined by the item reported by the household member (respondent)

Comment: Page 6: While sample size is outlined, how are these samples distributed across income and social groups, employment groups, insured, non-insured population, etc. must be described. The magnitude of OOP estimates and its variation are often influenced by these factors.

Response: Table 1 provides some demographic characteristics including, gender of household head, Marital status of household head, religion, age of household head and level of education: roughly half had no education. The main aim of the study was mainly to estimate and quantify the level of agreements between household reported expenditures and provider records regardless of the magnitude of the expenditure. The results do not suggest that these factors were different across the three questionnaire versions and therefore we do not expect our results to be influenced by these factors.

Comment: Page 7: the section on Health Financing System in Ghana must figure in the introductory section for readers to appreciate the current health financing mechanism in Ghana. IN any case, health financing system section cannot be clubbed with methods section.

Response: We have moved this section and renamed it. We found it a tricky section to place. We tried the Introduction, but it did not fit well there since it does not lead directly to the background of the study or the study aims. We have put it under the study settings section, and renamed it as Study context: Ghana's National Health Insurance Scheme.

Comment: Page 9: Ideally, the number of households interviewed should be considered as total sample rather than total sample identified as 1100 in each version. Or at least, please explain the reason for the difference between 1100 sample identified for each version and actual number of households interviewed. Also, it is not clear how many unmatched samples were drawn from the population and whether the sample size is robust needs to be described.

Response: the 1100 represents the target sample size (1000 plus 100, 10% non-response) per questionnaire version. However, due to non-response and some field challenges, the target sample size was not achieved for only version-1 of the study. We have now made this clear in the manuscript. Also, matching was not considered at the sampling stage and thus all samples were unmatched, matching was only done during data collection by immediately identifying the health care providers where the expenses were incurred and the corresponding provider information extracted to form a match record of households and provider data.

Comment: Page 12-13: As reported in the manuscript in page 12, the overall bias increases with increasing numbers of health items for OOPs in inpatient care and medicines. But table 4 supports this claim only for inpatient care, while overall bias for medicines and outpatient appear similar. Mean bias in both outpatient and medicines expenditure appear to slightly rise but with increase in number of items for medicines while for outpatient the number of items remain the same. This require urgent attention and edits in the manuscript and the results need to be interpreted accordingly. The difference in mean bias in preventive care services with similar items (5 items) also require explanation.

Response: We have revised the wording to make it clear that while the estimates in Table 4 are in the direction of increased for most of the estimates but significant only for inpatient and medicines. The effect of questionnaire on preventive care is not significant in Table 4. This has been made clear in the manuscript

Specific points to be addressed in the manuscript:

Comment: Page 4: Please expand COICOP, as it was introduced for the first time in the manuscript.

Response: This has been addressed in the manuscript now (Line 144)

Comment: Page 5: second line 'norther' should read as 'northern'.

Response: This has been addressed in the manuscript now (Line 151)

Comment: Page 7: Please check second line in Data Analysis section. The sentence has repeated twice 'reported OOPs' and 'reported OOPs'.

Response: This has been addressed in the manuscript now

Comment: Page 14: 5th Line: "preventing services" must be replaced by "preventive services". In last line 4 from bottom, 'provide' records must be replaced by 'provider' records.

Response: This has been addressed in the manuscript now

Reviewer: 1

Competing interests of Reviewer: None declared

Reviewer: 2

Competing interests of Reviewer: 'None Declared'.

We thank the reviewers for their insightful comments, suggestions and queries

VERSION 2 – REVIEW

REVIEWER	Selvaraj, Sakthivel Public Health Foundation of India, Health Economics
REVIEW RETURNED	30-Mar-2021
GENERAL COMMENTS	The authors have addressed all the questions and I am convinced that the manuscript is in a robust shape for publication.